# Non-pharmaceutical interventions in complementary and alternative medicine for insomnia in breast cancer survivors: a protocol for a systematic review and network meta-analysis

Qi Jin [1,2] Lumin Liu [1] Yuelai Chen,[1] Ping Yin [1]

QJ and LL contributed equally.

[1]Sleep Medicine Center, LongHua Hospital Shanghai University of Traditional Chinese Medicine, Shanghai, China
[2]Acupuncture Department, Shanghai Municipal Hospital of Traditional Chinese Medicine, Shanghai, China

**Correspondence to**
Mrs Ping Yin;
bingxue616@163.com and
Dr Yuelai Chen;
chenyuelai@163.com

## ABSTRACT

**Introduction** Insomnia has a high incidence in patients with breast cancer, which not only affects the quality of life of patients, but also affects the efficiency of later treatment and rehabilitation. Although the sedative and hypnotic drugs commonly used in clinical practice have a rapid onset of action, they are also accompanied by different degrees of sequelae, withdrawal effects and dependence and addiction. Complementary and alternative medicine (CAM) or complementary and integrative medicine, complementary integrative therapies, including natural nutritional supplement therapy, psychotherapy, physical and mental exercise, physiotherapy, have been reported to be used to treat cancer-related insomnia. Its clinical results are increasingly recognised and accepted by patients. However, the effectiveness and safety of these CAM are uneven, and there is no standard clinical application method. Therefore, in order to objectively evaluate the effects of different non-pharmaceutical interventions in CAM on insomnia, we will conduct a network meta-analysis (NMA) to explore the effects of different CAM interventions on improving sleep quality in patients with breast cancer.

**Methods and analysis** We will search all Chinese and English databases from the inception of the database to 31 December 2022. The databases include PubMed, Medline, Embase, Web of Science and Cochrane Central Register of Controlled Trials and the Chinese literature databases include CBM, CNKI, VIP, WANFANG. The Insomnia Severity Index and the Pittsburgh Sleep Quality Index will be considered as the primary outcomes in the study. STATA V.15.0 software will be used to conduct pairwise meta-analysis and NMA. Finally, we will use the recommended risk assessment tool RoB2 for risk and bias assessment, and use the Grading of Recommendations Assessment, Development and Evaluation evaluation method to evaluate the quality of evidence.

**Ethics and dissemination** Ethical approval will not be required because the study will not include the original information of participants. The results will be published in a peer-reviewed journal or disseminated in relevant conferences.

**PROSPERO registration number** CRD42022382602.

## STRENGTHS AND LIMITATIONS OF THIS STUDY

⇒ This study comprehensively compares the efficacy and safety of different non-pharmacological interventions for insomnia in breast cancer survivors to rank the superior efficacy of different interventions using network meta-analysis.

⇒ The quality of evidence will be assessed by the Grading of Recommendations Assessment, Development and Evaluation system and the risk and bias assessment will be done by risk of bias tool.

⇒ We will perform subgroup analyses based on different types of non-pharmacological interventions and conduct a meta-regression analysis with participants who have received or are receiving or have never received chemotherapy, radiation or surgery as covariates.

⇒ The study will focus on commonly used non-pharmacological interventions, such as cognitive–behavioural therapy for insomnia, acupuncture, yoga, tai chi, qigong, mindfulness therapy, physical exercise of different intensities (aerobic and resistance), tuina and massage, which may lead to limitations to application of the findings for clinical guidance.

⇒ We will only retrieve data from databases in Mandarin and English, which may restrict the amount of data available or lead to language bias.

## INTRODUCTION

Insomnia, as one of the most prevalent complaints among cancer survivors, mainly manifests as difficulty in initiating and/or maintaining sleep, early morning awakening, which are associated with impaired performance and daytime well-being.[1] According to the epidemiological investigation, 36%–59% patients experienced insomnia during the cancer trajectories. Among them, rates were

greater in patients with breast cancer from 42% to 69%.[2] In a 5-year follow-up investigation, 38% disease-free breast cancer survivors reported that they still had sleep problems.[3] Insomnia in breast cancer survivors can be debilitating, resulting in adverse impact on patients' life quality and exacerbation on other symptoms, including pain, fatigue, hot flash and mental disorders.[4 5] Moreover, insomnia was associated with poor survival rate for patients with breast cancer.[6 7]

Currently, the interventions in treating breast cancer-related insomnia include pharmaceutical therapy and non-pharmaceutical ones. Due to the drug adverse effects, as well as the psychological and physical dependence, complementary and alternative medicine (CAM) has begun to receive more and more attention. According to the definition of CAM by National Center for Complementary and Integrative Health (NCCIH), it is also called CIM (complementary and integrative medicine) or CIT (complementary integrative therapies), which includes four categories of treatment methods—natural nutrition supplements(such as dietary supplements, herbs and probiotics), psychology (such as mindfulness, meditation)), physical therapy (such as massage, spinal massage), a combination of psychosomatic exercises (such as yoga, tai chi, qigong, acupuncture) or psychological and nutritional (such as mindfulness diet).[8] Patients with cancer often use CAM therapy with recent estimates as high as 69%.[9] Most CAM methods, such as yoga, tai chi, meditation and music therapy, are safe for most patients. However, the use of herbs and supplements to treat sleep disturbance requires the supervision of healthcare professionals to prevent adverse reactions, interactions and toxicity.[10–12] Our network meta-analysis (NMA) fully considered the compliance and safety of CAM in the treatment of insomnia in patients with breast cancer. Therefore, only non-pharmaceutical intervention methods in CAM were retrieved, including psychotherapy (cognitive–behavioural therapy for insomnia (CBT-I), mindfulness, meditation, psychological education guidance), psychosomatic exercise (yoga, tai chi, qigong, acupuncture), massage, tuina and different intensity physical exercise. The purpose is to compare the differences and safety of non-pharmaceutical intervention in CAM in the treatment of insomnia in breast cancer, and offer some guidance and reference in clinical treatment.

CBT-I has been demonstrated as an effective treatment and widely recommended for breast cancer survivors.[13 14] Evidence shows that acupuncture can improve sleep quality with mental and emotional distress symptoms.[15 16] Mind–body therapies including yoga, taichi or qigong also seem to be available for breast cancer-related insomnia.[17 18] In addition, several studies reported that physical exercise can relieve the insomnia disorder of patients with breast cancer.[19 20] Tuina, a non-pharmacological intervention based on TCM Zang-Fu organ and meridian theories, has also been proven to help improve insomnia.[21] In conclusion, previous pairwise meta-analysis focused on only one CAM intervention mentioned above has demonstrated the efficacy in breast cancer related to insomnia.

However, apparent uncertainty about which non-pharmaceutical intervention is the most effective option makes it difficult for clinicians to manage this troublesome situation. In a randomised controlled trail (RCT), CBT-I showed superiority in shortening sleep-onset latency, reducing wake, improving sleep efficiency while acupuncture was better at increasing total sleep time (TST). In addition, the differences between acupuncture and CBT-I for women, non-whites, lower education and those with pain were insignificant under subgroup analysis.[22] Thus, a comparison of these non-pharmaceutical interventions and the relative effectiveness between them represents an impending need for patients with breast cancer.

NMA, which is also named multiple treatments comparison meta-analysis, helps to combine the direct and indirect evidence across a network of treatments. Furthermore, NMA is able to provide the ranking of treatment options based on their effectiveness.[23] However, to the best of our knowledge, so far NMA has not been used to perform the above functions in treating breast cancer-related insomnia. Therefore, by the mean of systematic review (SR) and NMA, this study aims to compare the efficacy and evaluate the safety of different non-pharmaceutical CAM interventions including CBT-I, acupuncture, yoga, tai chi, qigong, mindfulness therapy, physical exercise of different intensities (aerobic and resistance), tuina and massage for breast cancer survivors with insomnia.

In previous studies, there was only one SR report on physical exercise and mind–body exercise in the treatment of insomnia in breast cancer, and there was no specific SR study on insomnia in patients with breast cancer for other CAM therapies.[17] Based on our knowledge, this may be the first SR and NMA study to compare the efficacy and safety of non-pharmaceutical complementary and alternative therapies in breast cancer insomnia survivors according to Insomnia Severity Index (ISI), Pittsburgh Sleep Quality Index (PSQI) and sleep-related outcomes recorded by polysomnography data, actigraphy data or sleep diaries. The results of the study will provide a ranking of different complementary alternative therapy interventions to assist patients, physicians and policy-makers in their decision-making.

## METHODS
### Study registration
This protocol was registered on PROSPERO (CRD42022382602) and was reported following the Preferred Reporting Items for Systematic Reviews and Meta-Analyses Protocols (PRISMA-P) statement guidelines (online supplemental file 1 for PRISMA-P checklist). The findings of this study will be presented following the checklist of items to include when reporting a SR involving a PRISMA-NMA.

## Inclusion criteria
### Types of studies
We will incorporate RCTs that include multiple and single branches designs, full-text journal publications, unpublished clinical trials with online results in Chinese and English, and RCTs conducted in any healthcare setting.

### Participants
Studies enrolled participants (18 years and older) who were diagnosed with breast cancer (stages 0–IV) and whose insomnia occurred after the breast cancer diagnosis. Patients who are receiving or have completed radiotherapy or chemotherapy or surgery and patients with breast cancer who have never received radiotherapy or chemotherapy or surgery. There will be no restrictions on race, nationality and education.

### Types of interventions
Non-pharmaceutical CAM interventions, as the main treatment, will be included, which will be limited to several aspects of the following four treatments. (1) psychological: CBT-I with at least two components of cognitive therapy, stimulus control, sleep hygiene (including sleep education), sleep restriction, relaxation training and other variants of CBT-I; mindfulness, meditation; (2) physical: massage, tuina, aerobic exercise, resistance exercise, walking and interventions increasing physical activities and (3) mind–body exercise (such as acupuncture, including acupuncture, electroacupuncture, auricular acupuncture, yoga, tai chi, qigong). Studies used single or multiple complementary alternative medical intervention(s) will be considered.

### Types of comparator(s)/control
The comparison group will be included usual care, a waiting control, placebo, sham acupuncture and sleep education.

### Primary outcomes
The ISI and the PSQI will be considered as the primary outcomes in the study.

The ISI consists of seven items to assess the nature and symptoms of the subject's sleep disorders. Questions address subjects' subjective ratings of sleep quality, including the severity of symptoms, subjects' satisfaction with their sleep patterns, the impact of the degree of insomnia on daily functioning, subjects' awareness of the impact of insomnia on themselves and their level of frustration due to their sleep disorders.[24]

PSQI instrument produces a global sleep quality score and seven specific component scores: quality, latency, duration, disturbance, habitual sleep efficiency, use of sleeping medications and daytime dysfunction. Global scores, ranging from 0 to 21 with higher scores, indicates poor sleep quality and high sleep disturbance.[25]

### Secondary outcomes
Sleep efficiency, sleep latency, wake-up time after sleep onset and TST derived from polysomnography data, actigraphy data or sleep diaries will be regarded as secondary outcome measures.[26] In addition, adverse events will be extracted from the included articles.

## Exclusion criteria
1. Studies that are duplicates.
2. Patients with other diseases that affect insomnia.
3. The design type is non-RCT, including conference paper, summary, abstracts, editorials, clinical observations, case studies, cohort studies, non-randomised trials, case–control studies, any non-experimental investigation (including cross-sectional and retrospective surveys).
4. We generally exclude studies involving various types of cancer unless they have subgroup data for breast cancer.
5. The full text of the study is not available.

Any study that meets one or more of the above criteria will not be considered.

## Search methods for identification of studies
We will search all Chinese and English databases from the inception of the database to 31 December, 2022. The English literature databases include PubMed, Medline, Embase, Web of Science and Cochrane Central Register of Controlled Trials and the Chinese literature databases include CBM, CNKI, VIP, WANFANG. The search keywords were focused on 'CAM', 'CIM', 'CIT', ' breast cancer ', ' insomnia ', ' acupuncture ', ' electroacupuncture ', ' auricular acupuncture ', ' cognitive behavioral therapy ', ' mind-body therapies ', 'mindfulness', 'meditation', 'tuina', 'masssage',' yoga ', ' tai chi ', ' qigong ', ' physical exercise ', ' resistance training ',' walking ',' randomized controlled trial '. The final search formula was obtained by searching for the subject words, subordinate words and free words. The reference list of all selected articles will be independently screened to identify additional studies missed in the initial search. Table 1 shows specific search strategies for Web of Science.

## Data collection and analysis
### Selection of studies
Two researchers (QJ and LL) will screen the literature independently and extract the data and cross-check them. If there are differences, these will be resolved through discussion or negotiation with third parties. In the literature screening, the title of the article will be first read. After excluding obviously unrelated literatures, the abstract and full text will be further read to determine whether it will be included. If necessary, the authors of the original studies will be contacted by email or telephone to obtain information that has not been identified but is important to this study. EndNote V.X9 will be used to manage retrieved studies and remove duplicates. The selection procedure is presented in a PRISMA flow chart (figure 1).

### Data extraction and management
Two researchers (QJ and LL) will independently extract data based on a predesigned form. The extracted data

   3

| Table 1 | Search strategy for the Web of Science |
|---|---|
| Number | Search items |
| 1 | TS=(breast neoplasms OR breast cancer OR breast tumor OR mammary cancer OR breast carcinoma) |
| 2 | TS=(sleep initiation and maintenance disorders OR insomnia) |
| 3 | TS=(complementary therapies OR complementary medicine OR alternative medicine OR alternative therapies OR integrative medicine |
| 4 | TS=(acupuncture OR electroacupuncture OR auricular acupuncture OR tuina OR massage) |
| 5 | TS=(cognitive behavioral therapy OR mindfulness OR meditation) |
| 6 | TS=(mind-body therapies) |
| 7 | TS=(yoga) |
| 8 | TS=(tai ji OR tai chi) |
| 9 | TS=(qigong) |
| 10 | TS=(exercise therapy OR physical exercise OR aerobic exercise) |
| 11 | TS=(resistance training) |
| 12 | TS=(walking) |
| 13 | TS=(randomized controlled trial OR controlled clinical trial OR randomized OR placebo OR clinical trials as topic OR randomly OR trial) |
| 14 | TS=(animals) |
| 15 | TS=(humans AND animals) |
| 16 | ((#13) NOT #14) NOT #15 |
| 17 | #3 OR #4 OR #5 OR #6 OR #7 OR #8 OR #9 OR #10 OR #11 OR #12 |
| 18 | #1 AND #2 AND #16 AND #17 |

will be as follows: (1) basic information included in the study: first author and year of publication; (2) research methods: research type, intervention measures, control measures, sample size; (3) concerned outcome indicators and outcome measurement data and (4) follow-up. The corresponding authors will be contacted for missing information. The two researchers will cross-check the data after completion of data extraction. The disagreements will be solved by the team discussion or consultation with the third researcher (YC).

## Quality assessment

Two independent reviewers (QJ and PY) will complete the risk-of-bias assessment and evaluation of evidence certainty.

We will use Version 2 of the Cochrane tool for assessing the risk of bias in randomised trials (RoB2) revised by the Cochrane Methodology Group in 2019.[27] The quality of the study will be evaluated in the following five areas: (1) bias in the randomisation process, (2) deviation from established interventions, (3) bias in missing outcome data, (4) bias in outcome measurement and (5) selective reporting of bias in results.

The Grading of Recommendations Assessment, Development and Evaluation System (GRADE) system will be used to grade the quality of the evidence for main outcomes.[28] Evidence quality will be rated 'high', 'moderate', 'low' or 'very low' according to the GRADE rating standards. The evidence quality of a specific study will be assessed according to the risk of bias, inconsistency, indirectness, imprecision and publication bias.

## Pairwise meta-analysis

The pairwise meta-analysis will be performed using STATA V.15.0. Conventional pairwise meta-analyses will be performed with random-effects model. Pooled OR with 95% CI will be used for the dichotomous variables. Standardised mean differences or weighted mean differences with 95% CI will be used for the continuous variables. The heterogeneity between trials will be estimated using $I^2$ statistics and p value. The values of 25%, 50% and 75% for the $I^2$ as indicative of low, moderate and high statistical heterogeneity, respectively. We will explore sources of heterogeneity by subgroup analysis and meta-regression.[29]

## Network meta-analysis

If enough data are available, we will conduct an NMA to compare the efficacy of different interventions and controls. The NMA will be conducted on both direct evidence and indirect evidence in a Bayesian framework using STATA V.15.0 software. The publication bias will be evaluated with the Egger's test and funnel plots if the number of studies exceeds. A two-tailed p <0.05 is considered as statistically significant.[30] In addition, a node-splitting model will also be used to assess local inconsistency at the network level, and a consistency or inconsistency model will be selected based on the results. If p<0.05, there was a statistically significant difference between the indirect and direct multiple treatment comparisons. To evaluate the convergence of the results, we will analyse the potential scale reduction factor (PSRF), where a PSRF value close to 1 indicates successful convergence.[31]

## Subgroup analysis, meta regression analysis and sensitivity analysis

Subgroup analysis and meta-regression analysis will be used to explore possible sources of heterogeneity and inconsistency. Subgroup analysis will be performed according to the different types of non-pharmaceutical interventions. We will conduct a meta-regression analysis with participants who have received or are receiving or have never received chemotherapy, radiation or surgery as covariates. Sensitivity analysis is carried out by the method of exclusion to verify the robustness of the results.

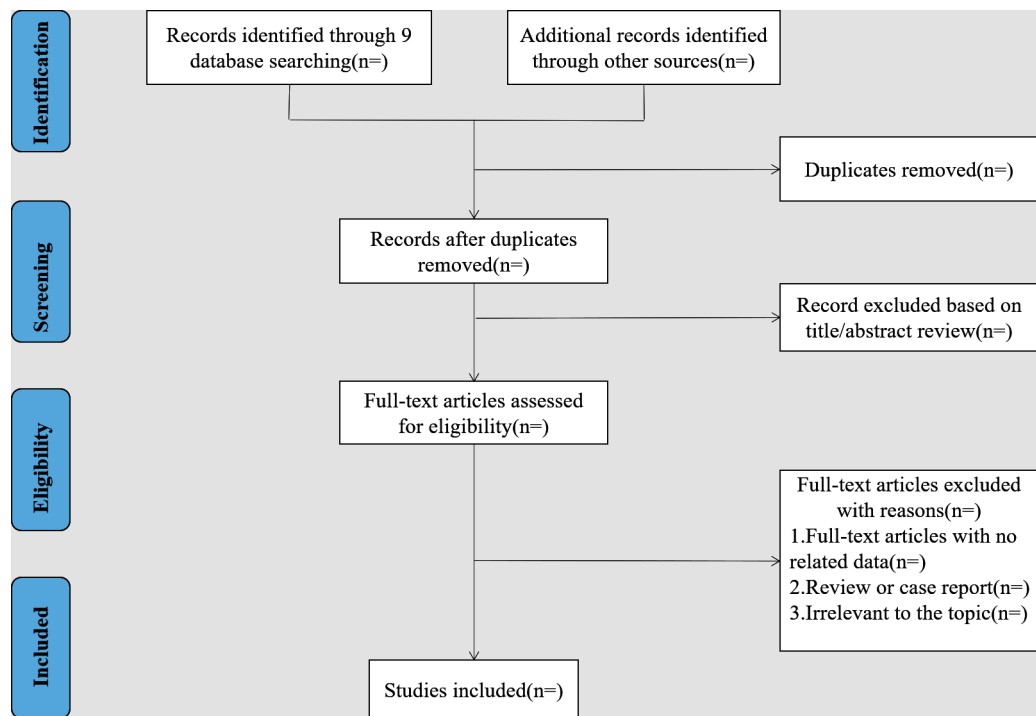

**Figure 1** PRISMA flow diagram of the study selection process. PRISMA, Preferred Reporting Items for Systematic Reviews and Meta-Analyses.

## Publication bias assessment

If more than 10 studies are included, a comparison-adjusted funnel plot will be generated to detect reporting bias.

## Patients and public involvement

This is an SR and NMA based on existing studies, so no patients or members of the public will directly be involved.

## Ethics and dissemination

The study will not require ethical approval because it comprises analysis based on existing studies. The results are expected to be published in a peer-reviewed journal or disseminated at relevant conferences. The objective of our study will evaluate the impact of complementary alternative therapies (including acupuncture, CBT-I, physical exercise, mind–body exercise, etc) on sleep-related outcomes in breast cancer survivors with insomnia by conducting an SR and NMA of existing RCT data for the purpose of providing reliable evidence-based medical evidence for complementary and alternative therapies for insomnia in breast cancer.

**Contributors** QJ and LL contributed equally to this article as cofirst authors. YC and PY contributed to the design of the study. The manuscript was drafted by QJ and LL. QJ and LL participated in the design of the search strategy and data extraction dataset. YC and PY will monitor each procedure of the review and be responsible for quality control. All the authors participated in the reading, discussion and revision of the manuscript, and all approved of the protocol for publication.

**Funding** This work was supported by the Clinical Incubation Program of the National Medical Center of LongHua Hospital to Shanghai University of Traditional Chinese Medicine, with grant number (GY202201).

**Competing interests** None declared.

**Patient and public involvement** Patients and/or the public were not involved in the design, or conduct, or reporting, or dissemination plans of this research.

**Patient consent for publication** Not applicable.

**Provenance and peer review** Not commissioned; externally peer reviewed.

**ORCID iDs**
Qi Jin http://orcid.org/0000-0001-7915-3007
Lumin Liu http://orcid.org/0000-0002-5981-2666
Ping Yin http://orcid.org/0000-0002-6379-4966

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
