## [Reviewer comments · BMJ Open]

ARTICLE DETAILS

TITLE (PROVISIONAL)	Nonpharmaceutical interventions in complementary and alternative medicine for insomnia in breast cancer survivors: a protocol for a systematic review and network meta-analysis
AUTHORS	Jin, Qi; LIU, LUMIN; Chen, Yue-lai; Yin, Ping

VERSION 1 – REVIEW

REVIEWER	Guo, Taipin Yunnan University of Chinese Medicine
REVIEW RETURNED	13-Feb-2023

GENERAL COMMENTS	1. The search keywords on complementary alternative therapies are not comprehensive and suggest to add.2. There is no distinction between sleep due to insomnia after breast cancer or insomnia itself and breast cancer later, suggest to revise.
---

REVIEWER	Narayanan, Santhosshi The University of Texas MD Anderson Cancer Center
REVIEW RETURNED	05-Mar-2023

GENERAL COMMENTS	Authors are covering an important topic of interest for breast cancer patients. The methodology and plan for this review is strong, and would be a nice addition to existing sleep literature when the review is complete. Please justify why you are including only since 2010. Also, add NIH/NCCIH definitions for CAM (Complementary and Alternative Medicine) therapy categories and how you chose the search terms (if based on NIH/NCCIH CAM therapy definitions, please include them). Also, the term CAM is changed to CIM(Complementary and Integrative therapies) recently. Please consider changing if relevant. Abstract: Well written abstract. However, the authors use some sentences which are not referenced in the manuscript. 1. Please add a reference for your sentence on how insomnia affects efficiency of later treatment and rehabilitation in the main manuscript or consider rewriting if no reference is found.2. I disagree with no standard application method. CAM therapies such as acupuncture, along with yoga etc. have been utilized in some academic cancer centers for managing insomnia. The following is one such paper describing the CAM therapies used by patients with sleep disturbances: Narayanan, S., A. Reddy, G. Lopez, W. Liu, S. Ali, E. Bruera, L. Cohen, and S. Yennurajalingam. 2021. 'Sleep disturbance in
--

	cancer patients referred to an ambulatory integrative oncology consultation', Journal of Supportive Care in Cancer. PMID: 34762218 Please consider changing to convey that despite effectiveness and safety of many CAM interventions such as acupuncture, it is only utilized in few cancer centers. Introduction: 3. I suggest taking the words drug resistance and adding drug adverse effects instead. 4. Please change the word "wildly" to widely 5. Please rewrite the sentence line 59, 60 page 4. Evidence shows acupuncture improves sleep quality..... 6. Page 5- lines 5 and 6- clarify which CAM intervention showed efficacy in breast cancer Methods: Overall, i agree with authors approach to methodology, however CAM includes natural products usage as well, such as melatonin, valerian etc. for insomnia. If you are not including them, please mention why you are not including them. If you are planning to include, then the relevant search terms needs to be added. I do see that you included this in limitations. If you are excluding natural products, then title should change to interventions, not complementary and alternative medicine, as CAM includes natural products too. 7. Are you planning to exclude patients who have not had chemotherapy or surgery, patients who refused treatment, patients who underwent radiation therapy. If not, please clarify as all breast cancer patients and survivors, in active treatment or surveillance. 8. Please include search terms- Mindfulness based stress reduction(MSBR), Oncology massage as search terms 9. Page 10-line 13- confidence interval? 20. Page 11- lines 5-8- How about surgery? Discussion: 21. Please review the reference, to check if association or causation. The claim that sleeping pills worsen the cancer may impact the patients who may decide not to use them, in order to not worsen the cancer, when they actually need it. I would be cautious in saying this, unless backed by strong evidence, more than one reference. Ethics: 22. Please double check with your institution as some requires a review and approval of systematic reviews as well, and include that no ethics approval was required by your institution as it comprises analysis based on existing studies.
--	---

VERSION 1 – AUTHOR RESPONSE

Reviewer: 1
Dr. Taipin Guo, Yunnan University of Chinese Medicine

Comment 1:
The search keywords on complementary alternative therapies are not comprehensive and suggest to add.

Response 1:
Thank you for your authoritative comment. We have added the complete definition of CAM, the

relevant search terms 'CAM', 'CIM', 'CIT', 'mindfulness', 'meditation', 'tuina', 'massage', and an explanation of why we have chosen the interventions we have included. (The revised section of the article is marked in red).

Comment 2:

There is no distinction between sleep due to insomnia after breast cancer or insomnia itself and breast cancer later, suggest to revise.

Response 2: Thank you for your professional advice. We have revised the patient inclusion criteria and defined insomnia occurs after a diagnosis of breast cancer. (The revised section of the article is marked in red).

Reviewer: 2

Dr. Santhosshi Narayanan, The University of Texas MD Anderson Cancer Center

Comment 1:

Please justify why you are including only since 2010.

Response 1:

Before drafting the plan, we roughly searched for relevant clinical treatment trials with "breast cancer" and "insomnia" as keywords, and preliminarily found that there were almost no relevant RCTS published before 2010, and only a few clinical observation articles. We have revised the retrieval time to all articles between the establishment of the database and 31 December, 2022. Thank you for your valuable suggestions.

Comment 2 :

NIH/NCCIH definitions for CAM (Complementary and Alternative Medicine) therapy categories and how you chose the search terms (if based on NIH/NCCIH CAM therapy definitions, please include them). Also, the term CAM is changed to CIM(Complementary and Integrative therapies) recently. Please consider changing if relevant.

Response 2:

Thanks for your advice. We have added treatment categories of complementary and alternative medicine in reference to the NIH/NCCIH definition of CAM, and explained why we included non-pharmacological CAM interventions in this study. (The revised section of the article is marked in red).

Comment 3 :

The part of Abstract

1.Please add a reference for your sentence on how insomnia affects efficiency of later treatment and rehabilitation in the main manuscript or consider rewriting if no reference is found.

2.I disagree with no standard application method. CAM therapies such as acupuncture, along with yoga etc. have been utilized in some academic cancer centers for managing insomnia. The following is one such paper describing the CAM therapies used by patients with sleep disturbances: Narayanan, S., A. Reddy, G. Lopez, W. Liu, S. Ali, E. Bruera, L. Cohen, and S. Yennurajalingam. 2021. 'Sleep disturbance in cancer patients referred to an ambulatory integrative oncology consultation', Journal of Supportive Care in Cancer.PMID: 34762218 Please consider changing to convey that despite effectiveness and safety of many CAM interventions such as acupuncture, it is only utilized in few cancer centers.

Response 3 :

1.Thanks to the expert's suggestion, we have added the corresponding reference. The description of this section in the manuscript is highlighted in yellow.

2.Thank you for the detailed review. We have reviewed the literature you recommended and reviewed the research progress of CAM in the treatment of cancer, especially breast cancer, redefined the treatment scope of CAM/CIT/CIM. The literature also suggested that "cancer patients often use complementary and integrative medicine (CIM) therapy. Recent estimates have been as high as 69%, with Sleep disturbance being a common factor in cancer patients wanting to receive CIM treatment

later. In the literature, we also know that acupuncture, yoga and other complementary therapies have been used by some academic cancer centers to treat insomnia, and have conducted standardized clinical trials. Therefore, we have revised this part and added corresponding reference. (The revised section of the article is marked in red).

Comment 4 :

The part of Introduction

1. I suggest taking the words drug resistance and adding drug adverse effects instead.
2. Please change the word "wildly" to widely.
3. Please rewrite the sentence line 59, 60 page 4. Evidence shows acupuncture improves sleep quality.....
4. Page 5- lines 5 and 6- clarify which CAM intervention showed efficacy in breast cancer.

Response 4:

Thank you very much for the expert's careful revision comments, we have made the wording in the corresponding position. The description of this part of the manuscript is highlighted in yellow.

Comment 5 :

The part of Method

1. Overall, I agree with authors approach to methodology, however CAM includes natural products usage as well, such as melatonin, valerian etc. for insomnia. If you are not including them, please mention why you are not including them. If you are planning to include, then the relevant search terms needs to be added. I do see that you included this in limitations. If you are excluding natural products, then title should change to interventions, not complementary and alternative medicine, as CAM includes natural products too.
2. Are you planning to exclude patients who have not had chemotherapy or surgery, patients who refused treatment, patients who underwent radiation therapy. If not, please clarify as all breast cancer patients and survivors, in active treatment or surveillance.
3. Please include search terms- Mindfulness based stress reduction(MSBR), Oncology massage as search terms.
4. Page 10-line 13- confidence interval?
5. Page 11- lines 5-8- How about surgery?

Response 5:

1. Thanks for your great suggestion on how to improve the professionalism of our manuscript. We have modified the intervention measures in the "Types of intervention measures" part of the manuscript, and changed the title of the article. The title will be modified as follows:
Nonpharmaceutical interventions in complementary and alternative medicine for insomnia in breast cancer survivors: a protocol for a systematic review and network meta-analysis (The revised section of the article is marked in red).
2. We do not exclude patients who did not receive chemotherapy or surgery as well as those who refused treatment and those who received radiation therapy, which has been revised in the article (The revised section of the article is marked in red).
3. We have added mindfulness, meditation, massage and tuina as search words to the manuscript and re-modify the search strategy. The description of this part of the manuscript is highlighted in yellow.
4. We have corrected the wording of the confidence interval correctly. The description of this part of the manuscript is highlighted in yellow.
5. We will conduct a meta-regression analysis with participants who have received or are receiving or have never received chemotherapy, radiation, or surgery as covariates. (The revised section of the article is marked in red).

Comment 6 :

The part of Discussion

1. Please review the reference, to check if association or causation. The claim that sleeping pills worsen the cancer may impact the patients who may decide not to use them, in order to not worsen the cancer, when they actually need it. I would be cautious in saying this, unless backed by strong evidence, more than one reference.

2. Please double check with your institution as some requires a review and approval of systematic reviews as well, and include that no ethics approval was required by your institution as it comprises analysis based on existing studies.

Response 6:

1. According to the editor's requirement of the magazine layout, we have deleted the section of Discussion.

2. We have confirmed with LongHua Hospital Shanghai University of Traditional Chinese Medicine that ethical approval is not required for systematic evaluation.

We would like to take this opportunity to thank you for all your time involved and this great opportunity for us to improve the manuscript. We hope you will find this revised version satisfactory.

Sincerely,
The Authors

VERSION 2 – REVIEW

REVIEWER	Narayanan, Santhosshi The University of Texas MD Anderson Cancer Center
REVIEW RETURNED	09-Apr-2023
GENERAL COMMENTS	All comments are addressed satisfactorily. Please consider writing what is tuina, as it is not a common term used in the west. I understand it is a type of massage. Thank you.

VERSION 2 – AUTHOR RESPONSE

Reviewer: 2

Dr. Santhosshi Narayanan, The University of Texas MD Anderson Cancer Center

Comment 1:

All comments are addressed satisfactorily. Please consider writing what is tuina, as it is not a common term used in the west. I understand it is a type of massage. Thank you.

Response 1 :

Thank you for your professional question. Tuina, a non-pharmacological intervention using fingers and strength, was developed from ancient therapeutic art and is slightly different from massage in Western medicine. Tuina is a treatment based on TCM Zang-Fu organ and meridian theories, and integrates modern scientific knowledge (such as biomechanical function, anatomy, pathology, and physiology) with traditional practice. In Chinese search terms, relevant literature of "tuina" may be involved, so we consider including the intervention method of "tuina". The following two references explain the concept of Tuina and its benefits, if you are interested in reading them. [1] Wang Z, Xu H, Zhou H, Lei Y, Yang L, Guo J, Wang Y, Zhou Y. A systematic review with meta-analysis: Traditional Chinese tuina therapy for insomnia. *Front Neurosci.* 2023 Jan 25;16:1096003. doi:

10.3389/fnins.2022.1096003. [2] Smith CA, Levett KM, Collins CT, Dahlen HG, Ee CC, Sukanuma M. Massage, reflexology and other manual methods for pain management in labour. *Cochrane Database Syst Rev.* 2018 Mar 28;3(3):CD009290. doi: 10.1002/14651858.CD009290.pub3.

The description of this section in the manuscript is **highlighted in yellow**. Below is a screenshot of the carefully edited article.

Cognitive behavioral therapy for insomnia (CBT-I) has been demonstrated as an effective treatment and widely recommended for breast cancer survivors.^{13 14} Evidence shows that acupuncture can improve sleep quality with mental and emotional distress symptoms.^{15 16} Mind-body therapies including yoga, taichi or qigong also seem to be available for breast cancer related insomnia.^{17 18} Besides, several studies reported that physical exercise can relieve the insomnia disorder of breast cancer patients.^{19 20} **Tuina, a non-pharmacological intervention based on TCM Zang-Fu organ and meridian theories, has also been proven to help improve insomnia**²¹ In conclusion, previous pairwise meta-analysis focused on only one CAM intervention mentioned above has demonstrated the efficacy in breast cancer related to insomnia.↵

Thank you very much for providing us with this opportunity to improve the manuscript. We hope you will be satisfied with the revised version.

Sincerely,

The Authors